

# Seven centuries of rainfall reconstructed from Scots Pine ring width in sub-Arctic Sweden

Petter Stridbeck[1], Jesper Björklund[1,2], Fredrik Charpentier Ljungqvist[3,4], Jennie Sandström[5], Mauricio Fuentes[1], Paul J. Krusic[3,4], Zhi-Bo Li[1], Kristina Seftigen[1,2]

[1]Gothenburg University Laboratory for Dendrochronology GULD, Department of Earth Sciences, University of Gothenburg, Gothenburg, Sweden
[2]DendroSciences, Swiss Federal Research Institute for Forest Snow and Landscape, Switzerland
[3]Department of History, Stockholm University, Stockholm, Sweden
[4]Bolin Centre for Climate Research, Stockholm University, Stockholm, Sweden
[5]Department of Natural Sciences, Mid Sweden University, Sundsvall, Sweden

*Correspondence to*: Petter Stridbeck (petter.stridbeck@gu.se)

**Abstract.** As the sub-Arctic and Arctic regions undergoes rapid changes, understanding its hydroclimate history is more critical than ever. However, limited availability of moisture-sensitive proxy data severely constrains our insights, underscoring the urgent need for more hydroclimate reconstructions in the region. Here we present a May–June precipitation reconstruction
based on ring width from living and dead trees of Scots Pine (*Pinus sylvestris* L.) growing under drought-stressed conditions at 63°N, near Skuleskogen National Park, on the northern part of the east coast of Sweden. The oldest deadwood sample dates back to the mid-11th century, and the Expressed Population Signal (EPS) exceeds 0.85 from 1320 CE until the present (2021 CE), making this the longest tree-ring-based hydroclimate reconstruction from high-latitude Fennoscandia. Unlike typical high-latitude forests in Fennoscandia, the trees at this site thrive under low-pressure conditions and show strong correlations
up to approximately $|r| = 0.6$ over the period 1920–2021 with drought-related variables such as precipitation, cloud cover, solar radiation, potential evapotranspiration and diurnal temperature range. The climate signal is concentrated to a short window between mid-May and early July but reflects climatic conditions over a broad region. Significant spatial correlations extend over most of Sweden as well as large parts of Norway and Finland, indicating sensitivity to large-scale climate systems. The tree-ring chronology also shows clear cyclic patterns, with a particularly strong ~64-year cycle. The modern era stands out for
its variability, with 2018 CE emerging as one of the of the driest year in the entire record. There is potential to extend the chronology further back in time, possibly to 1000 CE or earlier, and to extract additional climate information using other tree-ring parameters such as blue intensity.

## 1 Introduction

Dendroclimatologists not seldom journey to some of the world's most remote corners in search of untouched forests, free of
significant human influence, to extract an as pure climate signal as possible from the woods, and to be able to make long time-series (Schweingruber et al., 1988; Speer, 2010). They might search in sediments of lakes to potentially produce chronologies



spanning most of the Holocene (Grudd et al., 2002), seek out driftwood in the Arctic (Linderholm et al., 2021), or take samples from historical buildings (Labbas et al., 2024; Shumilov et al., 2025). Remarkably, a natural archive of living and dead *Pinus sylvestris* L., at the 'High Coast' on the northeast coast of Sweden, have been preserved for a millennium—possibly even
longer—on relatively modest hills, which rise a few hundred meters at the most, and which have been surrounded by human settlements since pre-historic times. Additionally, these trees give a distinctly different type of climatic information from what normally can be expected at high latitude forests. While tree-ring-derived chronologies from northern Fennoscandia have been an excellent source of past summer temperature proxy data (e.g., Melvin & Briffa 2013; Matskovsky & Helama, 2014; Linderholm & Gunnarson 2019; Fuentes et al., 2017; Stridbeck et al., 2022; Björklund et al. 2023), as they typically respond
strongly to growing season temperatures (Linderholm et al., 2015), the trees on the rocky terrain of the High Coast are instead primarily limited by moisture availability (Jönsson and Nilsson, 2009; Sandström et al., 2020). This offers a rare opportunity to reconstruct past hydroclimate conditions from a region where temperature has traditionally dominated the tree-ring signal. Recent advances in dendroclimatology have demonstrated the feasibility to produce long tree-ring based reconstructions of hydroclimate (precipitation or various drought indices) from regions with a relatively cold, and in some cases also a rather
humid, climate (e.g., Cook et al., 2015, 2020; Seftigen et al., 2015, 2017, 2020, 2025, Bebchuk et al., 2025). These reconstructions, derived from trees growing in unique micro-site conditions in otherwise relatively humid and cold locations (Ljungqvist et al., 2020), can carry a significant, time stable drought and/or precipitation signal almost comparable in strength to that of reconstructions from the arid and semi-arid regions of western United States that typically have the strongest hydroclimate signal of all tree-ring based reconstructions (for the latter, see St. George, 2014; St. George and Ault, 2014). In
Europe, north of the Mediterranean region, a number of millennium-long hydroclimate reconstructions from tree-ring data have been developed during the past 15 years. This includes tree-ring width based precipitation reconstructions from Central Europe (Büntgen et al., 2011) and various parts of Germany (Büntgen et al., 2010a; Land et al., 2019; Scharnweber et al., 2019; Muigg et al., 2020), a tree-ring width based groundwater reconstruction from the Upper Rhine Valley of southwestern Germany and northeastern France (Tegel et al., 2020), two tree-ring width based precipitation reconstructions from England
(Cooper et al., 2013; Wilson et al., 2013), and a May–June precipitation reconstruction from south-eastern Finland (Helama et al., 2003). In addition, numerous shorter tree-ring based precipitation reconstructions, including such based on isotope data, from temperate Europe have been published (e.g., Wilson et al., 2005; Labuhn et al., 2016; Loader et al., 2019).

Few high-resolution hydroclimate reconstructions exist for the sub-Arctic region (Ljungqvist et al., 2016), and the potential to develop tree-ring based such reconstructions for this cold region have been deemed challenging (Linderholm et al., 2018).
However, a large-scale assessment of the June–July climate signal in tree-ring width series across Eurasia north of 60°N by Hellmann et al. (2016) revealed that—in particular in regions in Siberia with a continental climate—a drought signal, rather than a temperature signal, prevailed up to 65°N at many locations. This included low-elevation sites in boreal Sweden and Finland (see Fig. 3 in Hellmann et al., 2016), corroborating earlier findings by Seftigen et al. (2015b). Similarly, Jack pine (*Pinus banksiana*) tree-ring chronologies developed from trees growing on rocky outcrop locations in the Northwest
Territories, Canada, (62.45°N, 114.37°W), have been found to contain a significant May–August precipitation signal despite



their sub-Arctic location (Pisaric et al., 2009). In summary, these studies provide evidence that hydroclimate-sensitive tree-ring chronologies can be developed from trees growing at high-latitude sites characterized by high solar radiation, relatively elevated summer temperatures, well-drained soils, and dry microsite conditions.

Seftigen (2015a) has previously developed a rather dense network of tree-ring width chronologies from *Pinus Sylvestris* L. collected at moisture-limited sites throughout southern Sweden. These typically exhibit a significant early growing season precipitation signal. In well-drained settings, it has been feasible even at Sweden's relatively precipitation-rich west coast. However, most of these sites are located towards the eastern generally drier parts of the country, including the island of Gotland, one of the regions with the lowest amount of precipitation in Sweden. In subsequent work, Seftigen et al. (2017) incorporated these and additional tree-ring chronologies from more northerly locations in Sweden, Finland, and Norway (extending up to approximately 63°N) to investigate large-scale hydroclimatic variability in Scandinavia over the past millennium. As part of this effort, they developed a reconstruction of the summer Standardized Precipitation-Evapotranspiration Index (SPEI) for a region encompassing roughly the southern half of Scandinavia. Importantly, this reconstruction was also employed to evaluate the ability of climate models to simulate variability in wet and dry conditions over the past millennium, well beyond the temporal scope of instrumental climate observations. The prospect to develop precipitation-sensitive tree-ring reconstructions much further north in Sweden, in the southern boreal zone, has been demonstrated by Jönsson and Nilsson (2009), who reconstructed May–June precipitation back to 1560 from Scots pine (*Pinus Sylvestris* L.) close to the town of Härnösand on the northeastern Swedish coast (63°N). By applying principal component regression to latewood (LW), earlywood (EW), and tree-ring width (RW) chronologies, they could explain 46% of the local variation in May–June precipitation. Sandström et al. (2020) developed a Scots pine tree-ring chronology from the Skuleskogen area (approximately 62.5°N–63.1°N, 17.9°E–18.7°E) to study the history of forest fires based on fire-scar evidence preserved in the trees. However, no climate reconstruction was undertaken as part of their study.

In this study, we present what is, to our knowledge, the longest northernmost tree-ring-based precipitation reconstruction in Northern Europe, based on Scots Pines (*Pinus sylvestris* L.) growing in a moisture-limited environment on the northeastern coast of Sweden at 63°N, near Skuleskogen National Park, approximately 60 km north of the town of Härnösand. The present work builds upon and complements previous studies, extending the spatial coverage of hydroclimatic reconstructions further north. Additionally, it enhances the temporal depth of such reconstructions beyond the past 300–400 years by incorporating a substantial amount of subfossil (deadwood) material, whereas earlier studies primarily relied on living trees.

## 2 Material and method

### 2.1 Study area

The study area is located just south of Skuleskogen National Park (63.0°N, 18.5°E) within the High Coast region (Sw. *Höga Kusten*), located on the northern part of Sweden's east coast (Fig. 1). This area is part of a distinctive and often dramatic landscape shaped by rapid post-glacial rebound following the retreat of the Weichselian ice sheet. The terrain is characterized



by steep cliffs, rocky outcrops interspersed with deep valleys and remains of raised beach terraces (Fig. 2). The region was heavily glaciated during the Last Glacial Period (*c.* 115,000 to *c.* 11,700 years ago), with an ice thickness reaching about 3000

metres. As the ice receded, the land began to rebound, causing an extensive uplift, which today still amounts to about 8 mm per year, making it the fastest rising landmass globally in present time. The highest coastline, located just west of the study site, reaches 286 metres above present sea level.

The soils in the study area are generally shallow and poorly developed, a consequence of both glacial scouring and subsequent wave erosion, which have exposed extensive areas of bedrock. These conditions create a moisture-limited environment despite

a moderate annual precipitation of 600–700 mm. Unlike most high-latitude forests in northern Fennoscandia, where temperature is typically the primary growth-limiting factor, the High Coast represents a rare example where tree growth is constrained by water availability.

Scots pine (*Pinus sylvestris* L.) dominates the tree cover, being well-suited to survive in these dry and nutrient-limited environments. Individual trees frequently reach ages of 300–400 years, with the oldest specimen found at the site exceeding

600 years. This exceptionally old tree, found growing in isolation on a raised beach terrace, may have survived longer than others due to reduced competition and some protection from the area's recurring forest fires. A notable feature of the site is the abundance of relict deadwood material, which can remain well-preserved on the ground for several centuries. This preservation allows for development of extended dendrochronological records that reach far beyond the lifespan of the oldest living specimen. The combination of poor soils, rocky terrain and frequent forest fires likely contributed to the limited historical

exploitation of these perched forests, despite their proximity to human settlements.

## 2.2 Sampling work

During 2021 and 2022, a reconnaissance survey of the area was conducted to investigate its potential for hydroclimate reconstruction. Sampling was subsequently carried out on Gårdberget, Vårdkallberget and Fanön located approximately 5 km

south of Skuleskogen National Park. Samples from living Scots pine trees were collected using 5- or 10-mm increment borers, sampling at breast height whenever possible. Very exposed areas on the mountain tops, where trees were twisted and bent, were generally avoided, as such individuals are likely to contain a high proportion of reaction wood. Similarly, relatively moist depressions were excluded from sampling. The dead wood of Scots pines in the area appear to remain well-preserved for extended periods after death, likely due to a high content of resin and the generally dry forest floor conditions. From the relic

material lying on the ground, samples were collected using a chainsaw, targeting sections with an estimated minimum of 50–100 years of intact growth rings.



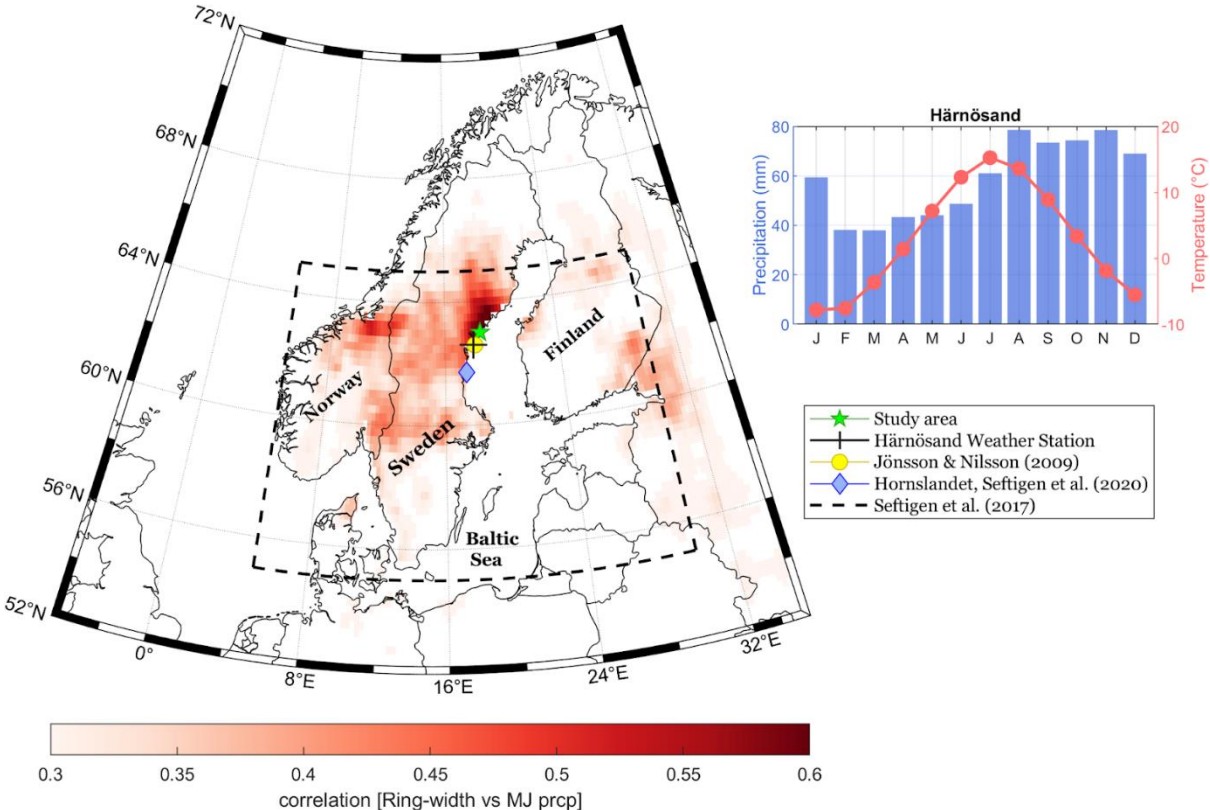

**Figure 1: Location of the study area, along with nearby moisture-sensitive tree-ring chronology sites. The dashed rectangle indicates**
**the area of a Standardized Precipitation-Evapotranspiration Index (SPEI) reconstruction conducted by Seftigen et al. (2017). The figure also includes the spatial correlation between the chronology presented in this study and May–June precipitation from the E-OBS gridded dataset (Cornes et al., 2018).**





**Figure 2: (A–C)** Typical terrain on the higher parts of the sampling area, with thin soil cover where mainly Scots pine trees thrive. **(D)** Raised single field formed by wave action when the shoreline was at higher level, now located about 50–100 meters above the present sea level. The pines that grow here can be relatively large and reach a remarkable age. Here the trees tend to grow in patches, which likely helps protect them from the recurring fires evident throughout the area. The oldest living tree here is well over 600 years old. **(E)** A small pine tree with thick roots that run across the bedrock to moister terrain. In this rocky terrain, it is not uncommon to see exposed roots nearly as thick as the stem itself. **(F)** Fire scars are often easily visible on both living trees and deadwood. **(G)** In one of the valleys just below the site, soil conditions are much wetter, supporting a relatively fast-growing spruce-dominated forest. **(H)** Example of deadwood found on the raised beach. In this dry environment, wood from pine can remain well-preserved for hundreds of years after falling. The sapwood tends to disintegrate relatively quickly, and in the older specimens, parts of the heartwood can also be missing.





## 2.3 Sample preparations and ring width chronology development


The samples were washed in 99% ethanol using a Soxhlet extractor for at least 48 hours, or until they no longer discoloured the solvent. The deadwood samples were sawn into slices approximately 4 mm thick before washing. The samples were then sanded with progressively finer sandpaper up to 800 grit. Figure 3 shows a section cut from a disk of deadwood where the heartwood has remained in excellent condition despite many centuries on the ground. Note the abundance of resin ducts, which

is typical of the pine wood from this site. Images of the samples were taken using the Skippy image-capturing system (Gärtner et al., 2024), integrated with a Sony Alpha 7R IV 61-megapixel camera, capable of producing images at resolutions up to 6000 dpi. At this resolution, well-prepared wood samples in good condition, allow for the distinction between cell walls and the lumen. Tree-ring width (TRW) measurements were conducted with CooRecorder (version 9.4; Larsson, 2014) and cross-dated using CDendro (Larsson, 2014) and COFECHA (Holmes, 2013). In addition to TRW, earlywood and latewood widths were

also measured. Although these parameters were not incorporated into the final precipitation reconstruction, they were retained for use in the climate–growth relationship analysis.

The extensive material collected allowed for a selective approached in choosing which samples to use; about half of the collected material has been incorporated into the chronology. The oldest samples from living trees date back to the 16th century. In some of the oldest, slowest-growing trees, the growth rings were often barely distinguishable even after thorough

sanding. In such cases, parts of the sample, or sometimes the entire sample, were omitted. Further, some rather long segments of deadwood have not been possible to date, suggesting that they could be older than the chronology presented here.

For standardization, a regional curve standardization (RCS) method (e.g., Briffa et al., 1992; Esper et al., 2003) was applied. Due to large differences in growth characteristics and growth rates among individual trees, an approach developed by Paul J. Krusic was used, multiple Regional Curve Standardization (march), which groups the samples into different age–growth

classes. It builds on the principle that fast-growing trees tend to die young and is offering a robust alternative to both RCS and single series detrending. The version we used partitions tree-ring series into up to 12 age–growth classes; two age classes; and six growth classes accounting for initial and mid-life growth relative to age cohorts, before applying regional curve standardization (methodological details are provided in Appendix A).

To evaluate the reconstruction skill of the mRCS chronology, a conventional calibration-verification approach was conducted

on it (Cook and Kairiukstis, 1990). The calibration period was split into two equally long segments and split-period calibration were performed in both directions. Model performance was assessed using correlations (r), reduction of error (RE), and coefficient of efficiency (CE). Further, the reliability of the chronology was assessed using the Expressed Population Signal (EPS), where values exceeding 0.85 conventionally are considered indicative of a sufficiently strong common signal for a robust climate reconstruction (Wigley et al., 1984). A linear model was then used to reconstruct total precipitation for May–

June. The uncertainty in the reconstructed time series was quantified using the root mean square error (RMSE) between the reconstructed and observed precipitation values. Uncertainty band was added to the reconstruction as ±RMSE which was smoothed with same spline as the reconstruction itself.





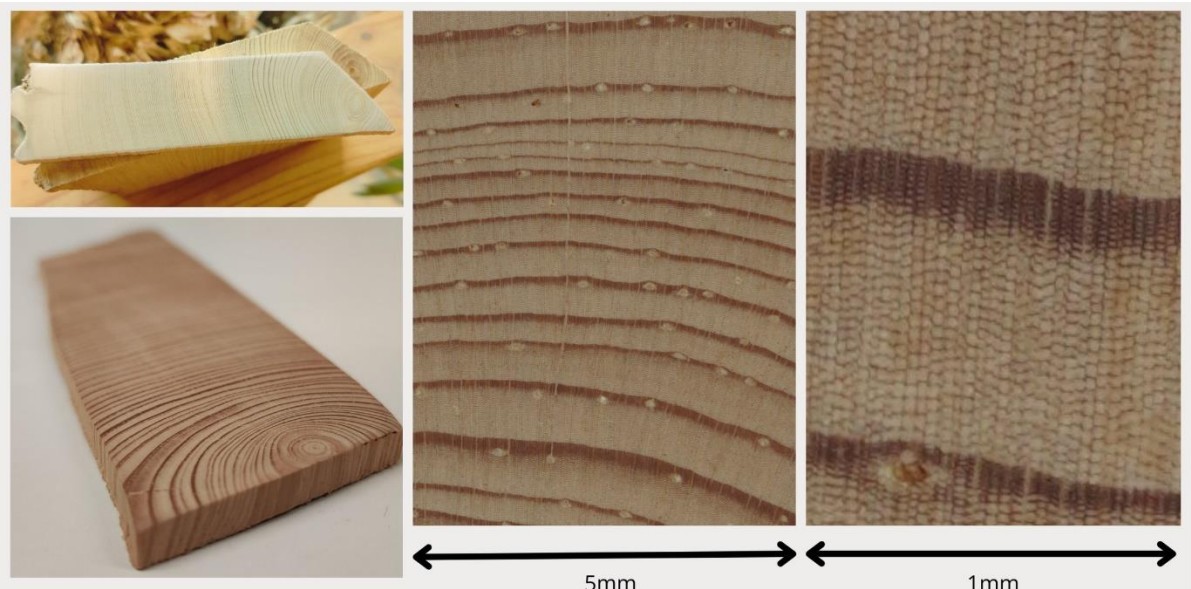


**Figure 3. Piece of dead wood dated to *c*. 1100–1350 CE, with the heartwood in very good condition.**

## 2.4 Meteorological data and power spectrum analysis

In this study, meteorological data from the town of Härnösand, located about 60 km south of the site were used. Additionally,
two grided datasets were used: the ensemble version of the E-OBS dataset (Cornes et al., 2018) and the CRU TS v4.08 dataset
(Harris et al., 2020). The E-OBS dataset provides temperature and precipitation at 0.1° × 0.1° resolution across Europe while
CRU offers global monthly climate data at 0.5° resolution. Incorporating both datasets allowed for including more variables
as well as examine differences in how the two datasets align with our chronology. For proper comparison, both datasets were
analysed over the common period of 1920–2021 and for an identical 1° × 1° region covering 62.5–63.5°N and 18–19°E.
Firstly, the climate–growth relationship was determined by correlating the mRCS chronology with various CRU variables as
well as E-OBS data for the current year. Encouraged by a noticeably stronger alignment between the precipitation data from
E-OBS and our chronology additional analysis was performed for this dataset incorporating for early- and latewood width as
well. For the spatial correlation analysis which is presented as maps covering the 52−72°N and 5−35°E regions, E-OBS was
used then available.
Lastly, precipitation data from Härnösand weather station was used, a record that starts already at 1859. However, it has some
missing entries up to and including the 1870s. In addition, there are disproportionately more extreme readings in the older part
of the record. One notable example that also falls within the target season is on the 18 June 1908, when the record state



precipitation of 187 mm, which is the 3<sup>rd</sup> highest 24-hour amounts ever reported in Sweden (SMHI, 2025). Although this may well be correct, a generally high level of extreme entries in the early part of the time series rises some suspicion about the quality of the older part of the record.

Nevertheless, excluding the earlier part of the time series, Härnösand was deemed to be most appropriate to use for the precipitation reconstruction. The correlation with the chronology was very similar to that of the E-OBS, however, a relatively low variability was identified in the first decades of the E-OBS dataset, which was not reflected in either the chronology or in the Härnösand time-series.

In order to explore and detect possible cyclicity in the precipitation reconstruction, we employed power spectral analysis using the NCAR Command Language (NCL). This is a technique used to examine the frequency components of a time series by decomposing its variance into different frequency bands, typically using Fourier Transform methods. Thus, it helps identify dominant periodicities and energy distributions within the data (Bingham et al., 1967).

## 3 Results

### 3.1 Ring width chronology

The mRCS ring-width chronology covers the period 1067–2021 CE. From 1526 CE onward, the sample depth remains consistently above 30 series. However, it gradually declines further back in time, and prior to 1320 CE, the EPS drops below 0.85 - the commonly accepted threshold for dendroclimatological reconstructions. A notable feature of the chronology is a strong decadal cyclicity, which is highlighted by the 30-year spline in Figure 4. Centennial-scale trends are less pronounced, though a general decline in growth from *c*. 1300 to 1700 is evident, followed by a tendency toward increasing growth from 1700 into the 20th century. In the modern era, the 1920s stand out with consistently high growth. In contrast, several recent years, particularly 2009 and 2018, show short-lived declines followed by quick recovery. The overall long-term trend over the from 1900 onwards remains relatively flat.

The distribution of all raw ring widths in the chronology (Fig. 4c) reveals a broad range of growth values with an upward-skewed distribution. Several hundred measured rings fall within the 50–100 µm range, while some exceed 1.5 mm at the upper extreme. The substantial variability among individual samples suggests tree stress, which may increase the potential to extract a strong climate signal. However, if trees with different growth rates are not evenly distributed over time, this can give rise to misleading trends in the time series, which the different growth classes in the mRCS standardisation aim to mitigate. See appendix A1 for the twelve age–growth classes the samples were sub-grouped into.





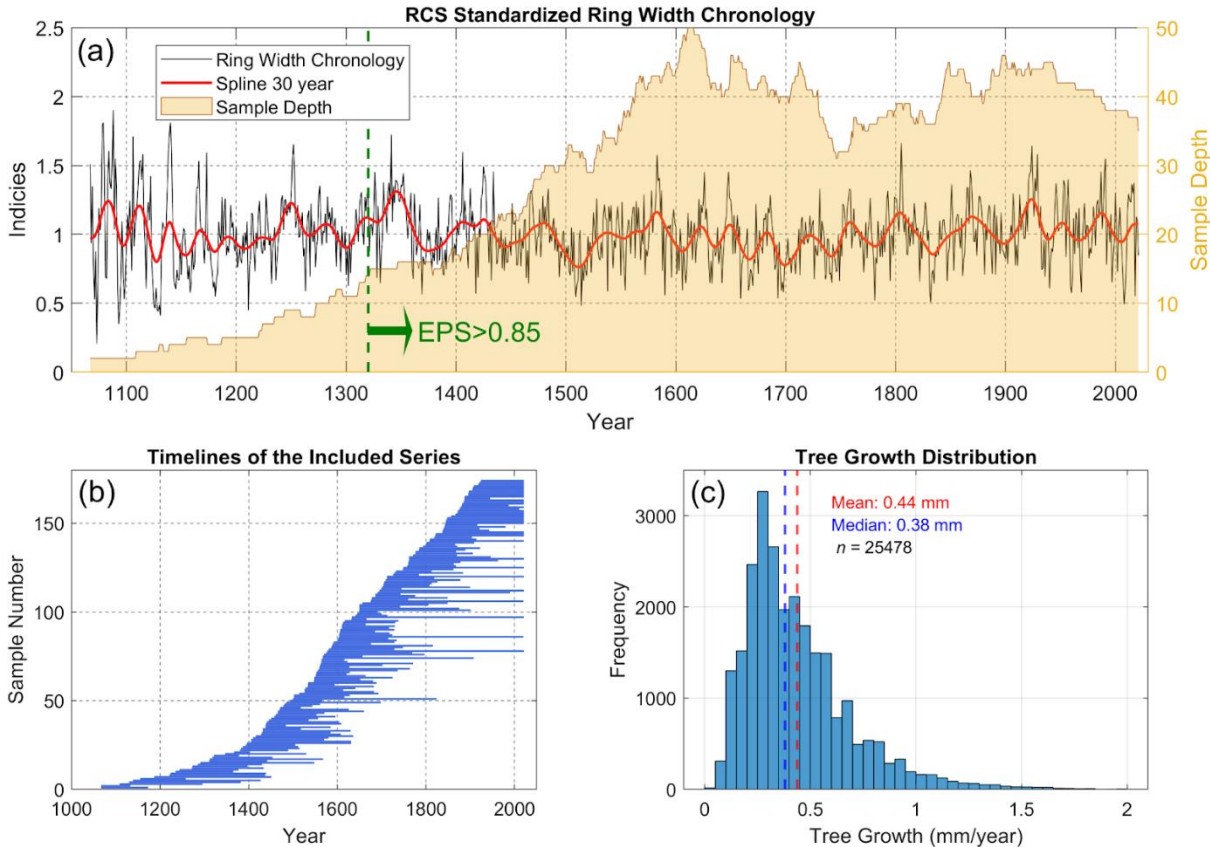

**Figure 4. Ring-width chronology built from 165 samples and 155 unique trees. (a) The tree-ring width chronology (black line), standardized using the multiple Regional Curve Standardization (mRCS) approach, smoothed with a 30-year cubic spline (red line). Sample depth is shown on the right y-axis. (b) The temporal extent of the individual series included in the chronology. (c) Distribution of all actual (non-standardized) tree-ring widths.**

## 3.2 Monthly correlations between precipitation and various climate variables

Figure 5 shows the monthly correlations between the ring-width mRCS chronology and various climatic variables from the CRU dataset. All variables exhibit statistically significant correlations with the chronology in June, with the strongest relationships found for diurnal temperature range (DTR), potential evapotranspiration, and the Standardized Precipitation-Evapotranspiration Index (SPEI). Variables associated with dry conditions consistently display negative correlations, and *vice versa*. While the CRU dataset offers a wide range of variables, precipitation data from E-OBS show substantially stronger correlations with tree-ring variables, whereas temperature correlations are similar for both datasets.

The E-OBS dataset (Fig. 6) shows considerably stronger correlations with the chronology, indicating a greater capacity to capture local variability in precipitation. When separating the growth ring into earlywood and latewood components, we see a tendency for later climate responses in the wood formed later in the season. September precipitation stands out with a





significant correlation, which is unexpected, as tree growth is typically expected to have ceased by that time. By the time of sampling in the second half of July 2022, the trees had, according to visual inspection, already developed a substantial portion of their latewood.

Mean temperature shows a negative association with ring width and is roughly the inverse of the precipitation signal, albeit weaker. The negative correlation between temperature and ring width during the growing season contrasts with what is frequently observed in relatively cool environments. However, it is noteworthy that periods of higher precipitation tend to coincide with lower mean temperatures, suggesting that the negative correlation with temperature may be an artifact of this inverse relationship. Diurnal temperature range (DTR), for which large values are indicative of clear weather and a lack of frontal precipitation, shows an almost inverse pattern to the precipitation signal during the critical May–June period.


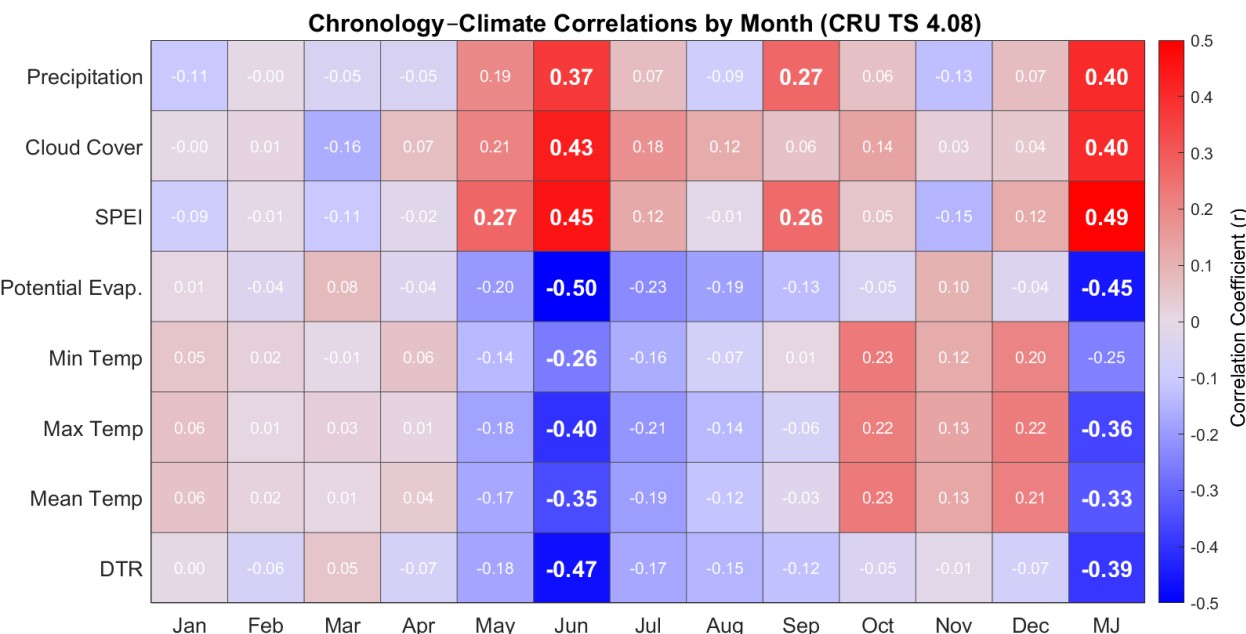

**Figure 5. Monthly Pearson correlations between various drought-related variables from the CRU TS 4.08 dataset and the ring-width chronology for the period 1920–2021 (except for global radiation, available from 1957–2021). The rightmost column shows**
**correlations with climate variables averaged over the May–June (MJ) period. All time-series were detrended using a 30-year spline before correlation analysis. Climate variables include precipitation, cloud cover, the Standardised Precipitation Evapotranspiration Index (SPEI), potential evapotranspiration, temperature, and diurnal temperature range (DTR). Statistically significant correlations ($P < 0.01$) are highlighted in bold.**






**Figure 6. Monthly Pearson correlations between the different ring width components and precipitation, mean temperature, diurnal temperature range using the E-OBS dataset for the 1920–2021 period. The ring width components are earlywood (EW) width, latewood (LW) width as well as the full ring width. Both the climate data and the ring width data have been detrended with a 30-years spline prior to correlation analysis.**

## 3.3 Running correlations: optimal target season

Figure 7 shows a running correlation analysis (31-year window) between tree-ring width and daily precipitation data from Härnösand meteorological station, calculated over 30-day moving periods. These periods are not confined to calendar months but can begin at any day of the year. For example, day 152 on the y-axis represents running correlation for the month of June. The precipitation signal remains generally stable in both timing and strength over the analysed period. The optimal target season—the window with the strongest correlation across the entire 1920–2021 period—falls within the May–June period





(May 7 to July 2, $r = 0.63$). A notable anomaly occurs between 1945 and 1975, when July shows a slightly negative correlation, due to a single outlier year. In that year (1960), a dry early summer was followed by heavy rainfall in July, yet tree growth remained relatively low (see Fig. 8). Excluding this outlier would shift the optimal target season further into July.

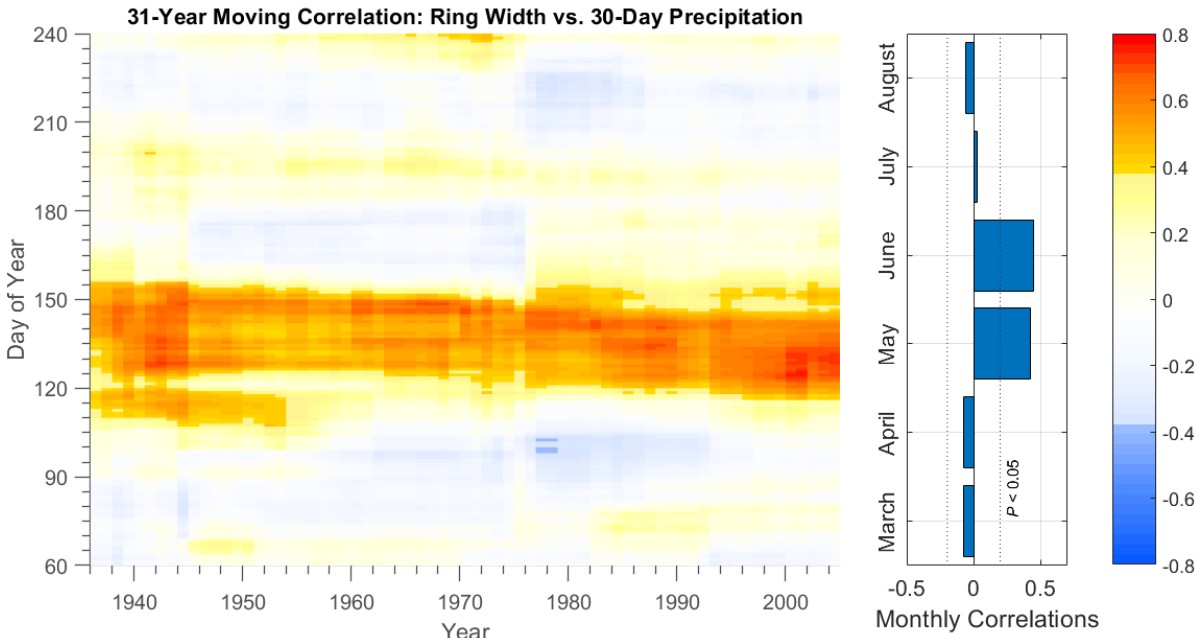

**Figure 7. Running 31-year correlation (Pearson) between tree-ring width and 30-day precipitation totals from the Härnösand meteorological station for the period 1920–2021. The y-axis (Day of Year) corresponds to the start date of each 30-day window. Insignificant correlations ($P > 0.05$) are displayed in pale colours. The bar chart displays monthly correlations for the entire 1920– 2021 period, the dotted line indicates a significant correlation for the entire 102-years period. The climate data and the chronology have been detrended with a 30-year spline prior to correlation analysis.**

### 3.4 May–June precipitation reconstruction

Based on the relationship between the ring-width index and instrumental precipitation data, May–June precipitation totals were identified as the most relevant period to reconstruct. Split-sample calibration–validation tests were conducted by dividing the periods 1880–2021 and 1920–2021 into two equally long parts (Fig. 8, see Appendix B for complete statistics). The latter period was chosen for the reconstruction due to quality concerns with the early part of the Härnösand record, which contains relatively extreme outliers. Figure 8b shows the average cumulative precipitation for 1880-2021, and three years were observed and predicted precipitation differ markedly. The dashed black lines represent the means of cumulative precipitation for the





years of upper and lower growth quartiles of the chronology, note that they are essentially identical until early May, then they diverge.

The model significantly predicts precipitation both forward and backward ($P < 0.01$), although it tends to underestimate the amplitude of the variability. As a robustness test Spearman's rank correlation ($\rho$) was tested in addition to the conventional Pearson's $r$. Spearman' $\rho$ yielded similar correlations (although consistently slightly lower), indicative of that the relationship

is linear and not driven by outliers.

Furthermore, although precipitation generally exhibits a skewed distribution, the May–June precipitation totals are only moderately positively skewed, and a Kolmogorov–Smirnov test (Massey, 1951) did not reject the null hypothesis of normality (see Appendix C, Fig. C1). Moreover, fitting non-linear models to the data produced trends that were essentially linear, with only negligible increases in explained variance (see Appendix C, Fig. C2). These results suggests that the simple linear

regression is a suitable model and that the conventional Pearson's $r$ is a suitable metric to describe its accuracy in our case. The linear regression model which was applied accounts for 35 % of the variance in May–June precipitation for the 1920–2021 period and was statistically significant at the $P < 0.01$ level (see Figure 8). The relationship was used to reconstruct the period 1067–2021, where the EPS values exceed the standard threshold of 0.85 from 1320 and onward (Fig. 9).

The spectral analysis indicates two dominant cycles in a close to 2:1 relationship, with maximum power at ~64 and ~34 years

(Fig. 10). To highlight these cycles, a 34-year and a 64-year spline are included in the figure 9. The wavelet analysis confirms that, for most of the time series, at least one of these cycles is significant ($P < 0.05$), with the exception of the 16[th] century, when a transition from the longer to the shorter cycle takes place. From 19[th] century onwards, the shorter cycle becomes dominant again.

A comparison with four existing hydroclimatic reconstructions for Fennoscandia (see Appendix D) shows that the two based

on Swedish tree-ring data—Jönsson & Nilsson (2009) and Seftigen et al. (2017)—show significant correlations with our reconstruction across common periods ($P < 0.01$), while the European gridded multi-proxy precipitation reconstruction by Pauling et al. (2006) and the Finnish tree-ring-based sunshine reconstruction by Gagen et al. (2011) show no significant relationship.






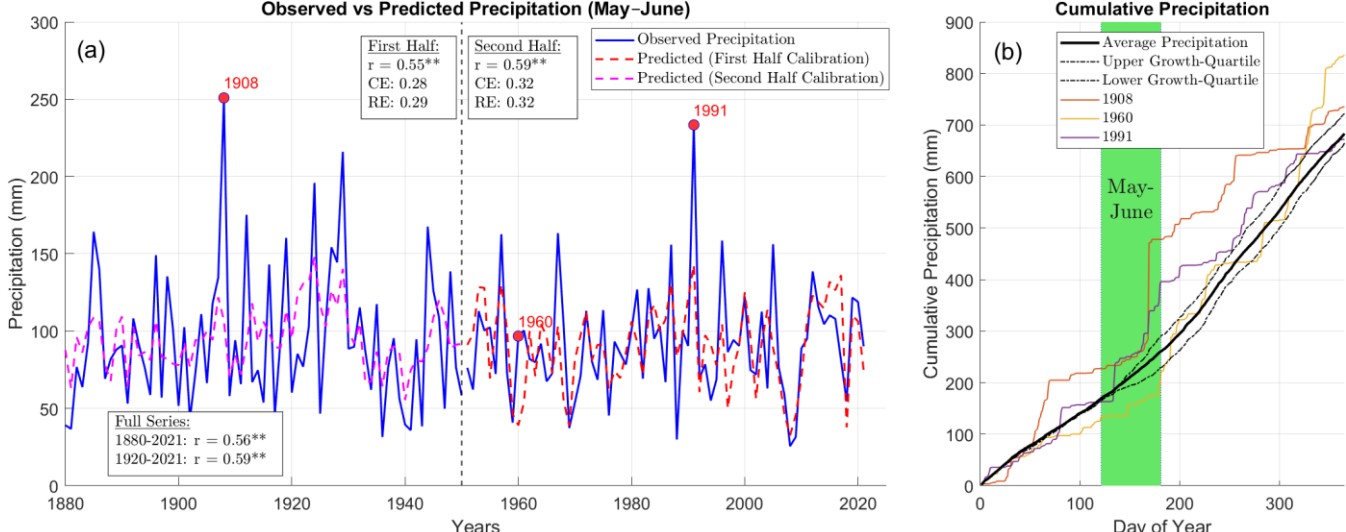

**Figure 8. (a) Calibration and validation of the model for May–June precipitation. The 1880–2021 period is divided into two segments (1880–1950 and 1951–2021), with linear regressions fitted to each period. The blue line represents observed precipitation, while the**

**dashed line shows reconstructed values based on the linear regression model from the other half period. The model's performance is evaluated using the Coefficient of Efficiency (CE), Reduction of Error (RE), and Pearson correlation (*r*) statistics between observed and modelled precipitation. (b). Mean cumulative precipitation over the year, alongside three specific years (1908, 1960 and 1991) where observed precipitation substantially exceeded model predictions. These years are characterized by dry spells during spring, with much of the May–June precipitation occurring in short, intense events. This is especially evident in 1908, when 187 mm of**

**precipitation was recorded on June 17th alone. Mean precipitation for years with fast and slow growth (corresponding to the upper and lower quartiles) is represented with dashed lines.**



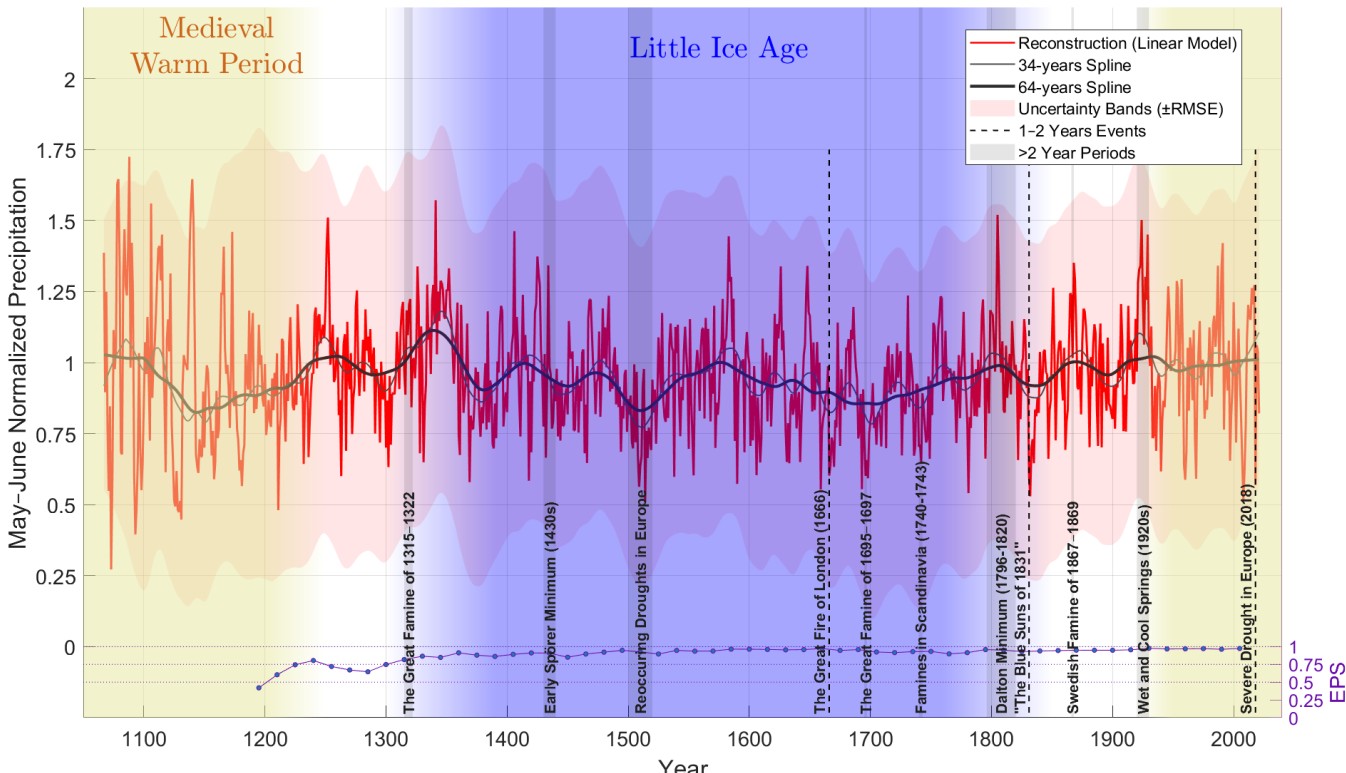


**Figure 9. Total May–June precipitation reconstruction based on the mRCS chronology, highlighting some significant historical events and periods of extreme weather or famine in Europe. A 34-years and 64-years spline represents the peak of the significant cycles ($P < 0.05$) identified in the power spectrum analysis. Below the chronology, the Expressed Population Signal (EPS) is plotted on the right y-**

**axis. Shaded area indicates ±RMSE uncertainty around the reconstruction**




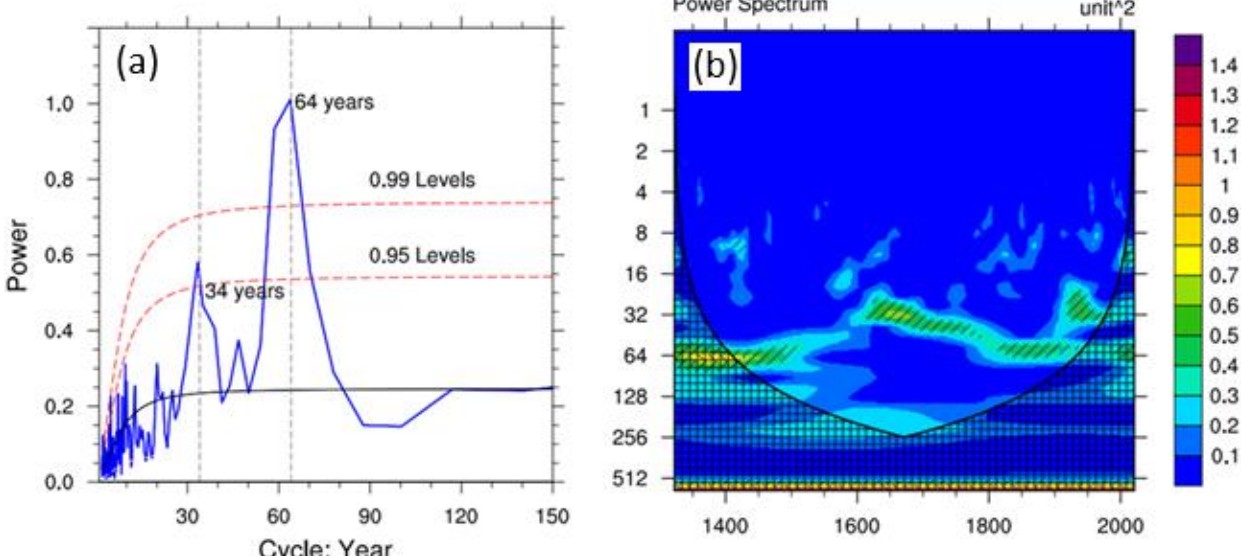

**Figure 10. Power spectrum and wavelet of the mRCS chronology over the period 1320–2021 during which the expressed population signal (EPS) remains above 0.85. (a) Peak years with significant power ($P < 0.05$) are indicated. Two dashed red lines denote the $P = 0.01$ and $P = 0.05$. (b) Wavelet indicating the stability of the cycles over time, significant regions are ($P < 0.05$) marked with diagonal**

**lines.**

### 3.5 Large-scale climate correlation patterns

Figure 11 shows the spatial correlation fields between the chronology and various climatic metrics—including precipitation, SPEI, cloud cover, solar radiation, and diurnal temperature range—over the May–June period. The upper panel displays results using the mRCS chronology, while the lower panel shows results based on the first-differenced chronology. Large fields with

significant correlations are primarily restricted to May and July, with July consistently showing the strongest individual-month signal (see Appendix G). Thus, the seasonal response window is short, but it explains, for some climatic parameters, conditions across most of Scandinavia.

Precipitation shows the strongest correlations locally, while the other parameters—particularly those involving a temperature component, such as SPEI and diurnal temperature range (DTR)—tend to exhibit large significant field. Applying first

differencing strengthens the precipitation correlation field, suggesting that high-frequency variability is better captured in the chronology than long-term trends. Interestingly, for all parameters except precipitation, the areas of strongest correlation tend to appear to south or southwest of the site (which is indicated by a red dot).




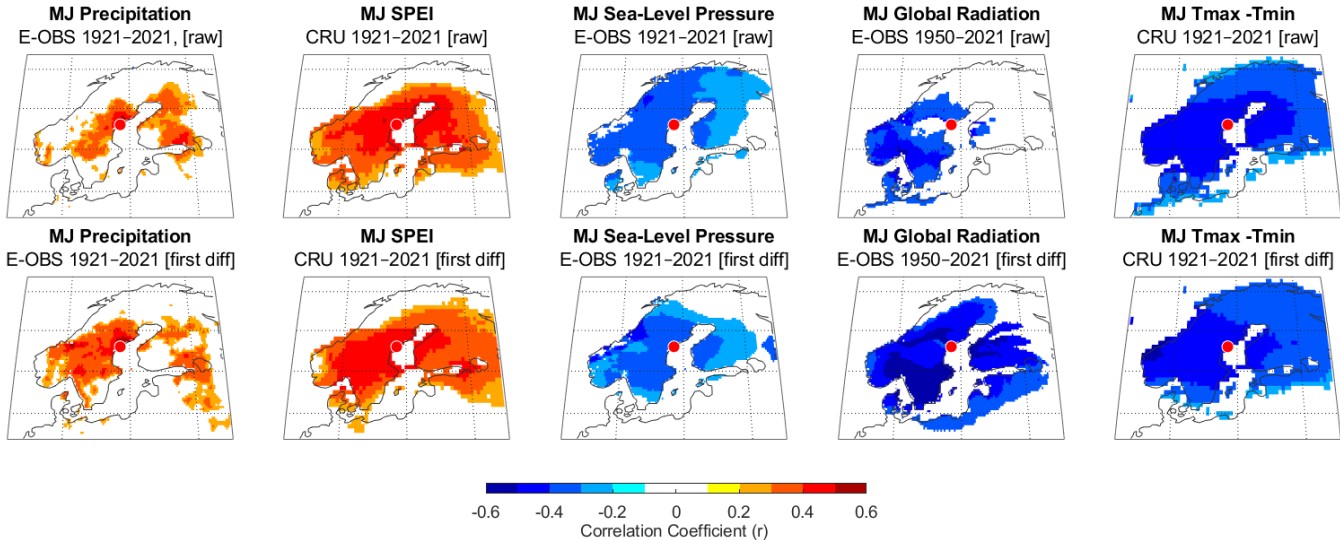


**Figure 11. Correlation maps (Pearson's r values, *P* < 0.01) between tree-ring width chronology and the instrumental data for May–June over the 1921–2021 period. Raw data (upper row) and first-difference data (lower row) are used. The red dot denotes the location of our sampling site.**




**4 Discussion**

Reconstructing hydroclimatic conditions can be more challenging than reconstructing temperature, partly because precipitation varies more locally and tends to have a skewed distribution (Büntgen et al., 2010b; Bunde et al., 2013; Franke et al., 2013;



Zhang et al., 2015). Here we have successfully developed a hydro-climatological reconstruction that extends beyond the spatial limits of most previous studies. In our view, this makes the study valuable, even though the relationship between rainfall and

tree growth is not particularly strong, and the reconstructed period only spans two months.

With regard to the site conditions, the relative lack of human influence at the site we do believe is related to the contrasting growing conditions compared to the surrounding valleys, which are fertile and moist, with much faster growing trees. Further, the shingle field at the lower part of Gårdberget–where the oldest living trees have been found–was the most accessible part of the site for sledges, draught animals, or similar means in historical times. This may have facilitated the transport of materials

from the area and could explain the relative scarcity of deadwood at this location.

## 4.1 The climate signal and associated uncertainties

The reconstruction explained a relatively modest 35 % of the precipitation variance, leaving substantial room for growth-limiting factors beyond total May–July precipitation. Given the large uncertainties in the reconstruction, identifying long-term

trends or influence from recent climate change is challenging. What we can state with high statistical confidence is the presence of a strong cyclical pattern in the record, although the exact origin of this pattern remains speculative.

For instance, the possibility that long-term trends partly are driven by temperature should not be dismissed, as indicated by relatively low growth during portions of the Little Ice Age (*c*. 1300–1850; Wanner et al., 2022). Nevertheless, chronology consistently showed significant responses to all drought-related variables examined for one or more months. And years with

unusual low growth ought to very likely correspond to dry conditions during the growing season. Importantly, whether rainfall occurs as intense showers or is spread out over time can greatly affects water availability, particularly in well-drained conditions as at this site. Examples of years with large amounts of precipitation concentrated in extreme events and/or with very low precipitation prior to the target season of May–June (as shown in Figure 8b) had significant deviations between observed and predicted precipitation.

A theme throughout this work is that the most fundamental methodologies for reconstructing climate from tree rings have been used (with the exception of the mRCS standardization), with the ethos to not add complexity if it do not yield meaningful improvements in model performance. Obviously, an overly simplified model may fail to capture much of the signal in the data, and it is reasonable to assume that the precipitation response is dampened at high precipitation levels. However, fitting non-linear regressions such as a second-degree polynomial or a power function, also resulted in close to linear trends and a

negligible increase in explained variance. Further, the linear regression calibration method used to estimate precipitation from the chronology tends to underestimate the amplitude of precipitation variability, particularly at low frequencies. However, alternative calibration methods, when accounting for the level of explained variance, would instead risk of overestimating the variability. To avoid underestimation (or overestimation) of low-frequency variability, a correlation coefficient of approximately $r = 0.7$ or higher would be required (Christiansen and Ljungqvist, 2017).



Our reconstruction, Jönsson & Nilsson. (2009), and Seftigen et al. (2017) all show significant commonalities over their overlapping periods ($P < 0.01$) (Appendix D1). A notable difference is that Jönsson's reconstruction has much less amplitude in the decadal cycles. The SPEI reconstruction by Seftigen et al. (2017) shares common cyclic patterns with ours but also shows periods of divergence, particularly during the 1800s. Noteworthy, it is to be expected that this reconstruction differs substantially, as it has a one-month later target season, cover a broader region, and is for a drought index rather than direct
precipitation.

The gridded seasonal precipitation reconstruction for Europe developed by Pauling et al. (2006), which covers the period 1500–1900, was based on a wide variety of instrumental series as well as different types of natural proxies, including tree-rings, but it lacked agreement with our reconstruction. It should be noted that no paleoclimate records from Fennoscandia were included in their work. Another interesting comparison is the more than a millennia long sunshine reconstruction from western
Finland by Gagen et al. (2011). However, it lies at the margin of the significant correlation field for global radiation of our chronology and year-by-year correlations show no significant relationship; however, long-term trends exhibit some tendency to be inverted as could be expected as our chronology have a significant negative correlation with sunshine for a large part of Fennoscandia. It is noteworthy that these reconstructions overlap with ours only for the month of June, as their target seasons are June–August and June–July, respectively.


## 4.2 Comparison of climate datasets

The CRU dataset appears capturing local variability in precipitation to a lesser extent than other records used for this study. The Standardized Precipitation-Evapotranspiration Index (SPEI) being the only CRU variable showing significant correlation with the chronology for both May and June as well as having a relatively large significant correlation field. That the CRU
derived SPEI locally exhibits weaker correlation with the chronology than E-OBS precipitation ($r = 0.49$ vs. $r = 0.61$) originates likely from the precipitation component in the index. It would be of great interest to explore a drought index based on datasets that better reflect year-to-year precipitation variability than CRU currently does in the region.

It has further been suggested that the more complex Palmer Drought Severity Index (PDSI) is more suitable for high-latitude regions with seasonal snowpack, such as Scandinavia (Raible et al., 2017). However, the PDSI responds slowly and reflects
conditions over the preceding 9–12 months, our site, by contrast, has thin, well-drained soils, and tree growth is affected by meteorological conditions over a much shorter period. This is evident from the fact that May is the first month of the year to show a positive correlation with precipitation. Moreover, Seftigen et al. (2017; see Appendix D) conducted an extensive evaluation of tree growth sensitivity to drought, applying the SPEI across multiple accumulation periods (1 to 24 months). While our specific site was not included, the study encompassed ecologically analogous sites across Scandinavia. Their
findings demonstrate a consistent and robust growth response to short-duration summer drought events, emphasizing the limited relevance of long-term drought indices such as the PDSI in such settings. These results suggest that snowpack and




associated meltwater inputs are of marginal hydrological significance in environments with limited water-holding capacity and rapid soil drainage.

Precipitation data from Härnösand weather station outperformed both E-OBS and CRU in terms of reflecting local conditions,
particularly prior to 1950–even though the station is located more than 50 km from the site. This may be due to the fact that both locations share similar environmental settings—they are coastal and separated from the open Baltic Sea by a few kilometres of archipelago.

## 4.3 Trends, cycles, historical events

Severe droughts across Europe over the past millennium are clearly imprinted in European tree-ring data (Cook et al., 2015). Droughts, sometimes lasting multiple years, were also occurring during portions of the Little Ice Age and are well documented in documentary data (Pfister and Wanner, 2021) and tree-ring data in Central Europe (Büntgen et al., 2021; Arosio et al., 2025). The sparse documentary data over climatic conditions in Sweden from the 16th century reveals dry conditions for most on that century (Leijonhufvud and Retsö, 2021), something not shown in the Skuleskogen tree-ring chronology. Such differences may
be related to geography as most documentary records derives from the southern one-third of Sweden (below about 60°N). Several historically documented droughts are traceable in the Skuleskogen tree-ring chronology, while others are not, likely because the study site is too distant to reflect the climate impacts experienced in more densely populated agricultural regions (for historical drought-impacts on agriculture in southern Sweden, see Skoglund, 2022, 2023).

Interestingly, some periods of reduced solar activity, typically connected with colder climate phases, such as the early phase
of the Spörer Minimum in (1430s), and the Dalton Minimum (1790–1830 CE), coincide with favourable growing conditions and rapid tree growth. This emphasises the drought-sensitivity of the Skuleskogen tree-ring chronology. However, a period of consistently low growth occurred about 1700 CE following the cold 1690s. This coincided with a period which had severe harvest failures across large portions of Europe (Ljungqvist et al., 2024) including the Nordic countries where it in the Swedish Empire (that included present-day Finland and Estonia) culminated in the great famine of 1695–1697 (Dribe et al., 2017). In
present-day Finland, it is estimated that as much as nearly one-third of the population died during this famine (Huhtamaa et al., 2022). This event represents a notable exception to the climate–growth relationship observed at our site. While cool summer conditions are generally favourable for growth here, the extreme severity of this period appears to have pushed conditions beyond a critical threshold, disrupting the usual link between climate and tree growth.

In contrast, the severe harvest failure in northern Sweden 1867 CE (Forsberg and Bohman, 2023), triggered by an unusually
cold spring and short summer, is not reflected by low values in the Skuleskogen tree-ring chronology. This highlights the exceptional nature of the minimum around 1700 CE, when cold conditions appear to have been so extreme that trees, which are typically limited by water availability, transitioned temporarily to be at least partly limited by temperature. Despite equally cold, or colder, summer conditions in Sweden during parts of the late 16th century and early 17th century (Linderholm et al.,




2015), a similar growth depression did not occur as the one around 1700 CE. The reasons for this, including the possible role
played by different hydroclimatic conditions, warrant further research.

Furthermore, although the reconstructed period—May to June—is relatively short, it is critically important for agriculture in Sweden (Skoglund, 2023). Our new precipitation reconstruction from Skuleskogen explains large-scale weather patterns not only near the site but for much of southern Sweden as well. This means that the Skuleskogen tree-ring chronology can provide valuable information on May–June hydroclimate conditions also in the southern one-third of Sweden where drought can be
expected to result in poorer harvest yields for most major crops historically (Edvinsson et al., 2009; Ljungqvist et al., 2023) as well as today (Skoglund, 2023). Thus, the new dataset presented here offers the potential to improve our understanding of past harvest variations over the past millennium.

A striking feature of the reconstruction is perhaps the strong multidecadal oscillation, there the strongest of the two significant cycles in the power spectrum peaks at ~64-years. An interesting observation—though no quantitative comparison has been
conducted in this work—is that oscillations with similar wavelengths (~60–80 years) have also been identified in the Atlantic Multidecadal Oscillation (Enfield et al., 2001) and the Pacific Decadal Oscillation (Mantua and Hare 2002).

## 4.4 Prospects for extending the chronology further back in time

About a dozen lakes in the area, both near the site and further inland, were investigated for subfossil material. However, the
available material appears to be limited and/or originates from environments with persistently moist conditions, making the woods from these sites less suitable for extending the existing chronology. Numerous small pieces of deadwood, potentially very old, remained undated in this study, and some longer segments characterized by non-complacent growth patterns were also undatable, suggesting that they may predate the established chronology. This presents a promising opportunity to extend the record further, contingent upon successful dating of additional material from the site. Radiocarbon ($^{14}$C) dating could
initially help verify whether these undatable samples are indeed older than the current reference chronology or simply lack synchronous growth signals necessary for cross dating. The long-term preservation of wood at this site, evidenced by well-preserved samples such as the one shown in Figure 3, supports the potential to extend the chronology beyond a millennium. This underscores the broader opportunity to develop long tree-ring-based hydroclimate reconstruction from carefully selected sites in the sub-Arctic boreal forest zone—a region that remain underutilized in high-latitude tree-ring-based hydroclimate
reconstructions. Looking ahead, we aim to analyse Blue Intensity data from the same material to further assess its potential for capturing hydroclimatic variability across high-latitude environments, with the aim of advancing large-scale climate reconstructions in regions where such records remain critically underrepresented.




## 5 Conclusions

We present the first tree-ring-based precipitation reconstruction from boreal sub-Arctic Sweden extending back to medieval time (with EPS > 0.85 from 1320 CE to 2021 CE). Drought related variables such as precipitation, SPEI, sunshine, cloudiness
and sea-level pressure significantly correlates with the chronology during a short but critical season typically ranging from late spring to early summer, with no or little climate signal beyond early July. Significant spatial correlations extend across a much of Fennoscandia for May–June. The broad spatial signal, particularly for the wetter southwestern parts of Scandinavia, demonstrates the added value of our dataset in regions where identifying drought-sensitive trees is inherently challenging. There is a strong potential to extend the current tree-ring chronology back to at least 1000 CE, if not earlier. Reconstructions
of other variables, such as global radiation or a drought index is also feasible, underscoring the broader opportunity to develop long tree-ring-based hydroclimate reconstruction from carefully selected sites in the sub-Arctic boreal forest zone.





## Appendix A. Description of the mRCS software (version 12)

What follows is an abridged description of the multiple Regional Curve Standardization (mRCS) programs that represent an attempt to marry the need for removing non-climatic biological trends with conserving low-frequency climatic trends in annual tree-ring increments. Version 12 of the mRCS can distinguish up to 12 age–growth classes; two age classes, short and long; and six growth classes. A first order growth class distinction is based on the performance of each series' initial growth relative to that of the mean or median of the cohort's initial growth. For example, in version v12 the initial growth rate of the two age

cohorts, Short and Long, is partitioned into three growth classes; ShortLow; series with an initial growth rate that is $<(\mu-1\sigma)$; ShortHigh; series' initial growth rate is $>(\mu-1\sigma)$; and ShortMid; all that remain. The next three growth conditions are based on the performance of the middle and final third of each series' growth rate. For example, the v12 age–growth class; ShortLow_LowMid contains all those series with a low initial growth rate, relative to the initial mean or median of all series in the short age cohort; and a low mid-life growth rate relative to the rate of growth in the series' final ~1/3$^{rd}$ measurements

(i.e., _LowMid). Conversely, the age–growth class LongHigh_HighMid describes long series that have overall high initial growth, and relatively high mid-life growth in relation to the growth rate in the last third of the series' length. In the end there are 12 age–growth classes.

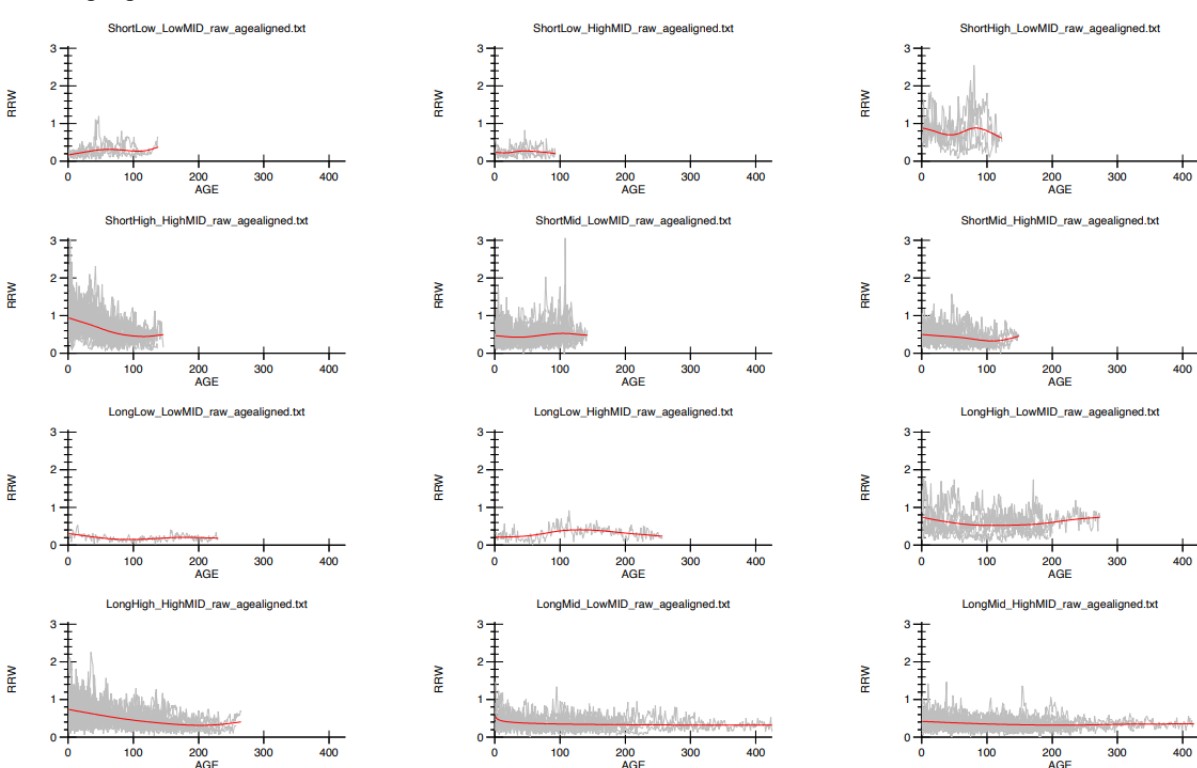

**Figure A1. The 12 age–growth classes which samples were sub-grouped into.**






**Appendix B. Statistics of calibration–validation tests**

**Table B1. Split-sample calibration–validation results for the relationship between May–June Härnöand precipitation totals and the mRCS ring-width chronology, evaluated over the periods 1880–2021 and 1920–2021.**

| Calibration | | | | Verification | | |
|---|---|---|---|---|---|---|
| Period | Pearson's r | $r^2$ | Spearman's ρ | Period | RE | CE |
| 1880–1950 | 0.55 | 0.30 | 0.53 | 1951–2021 | 0.32 | 0.32 |
| 1951–2021 | 0.59 | 0.35 | 0.57 | 1880–1950 | 0.29 | 0.28 |
| 1920–1970 | 0.56 | 0.31 | 0.48 | 1971–2021 | 0.40 | 0.40 |
| 1971–2021 | 0.64 | 0.41 | 0.63 | 1920–1970 | 0.30 | 0.30 |
| **1880–2021** | **0.56** | **0.31** | **0.54** | | | |
| **1920–2021** | 0.59 | 0.35 | **0.55** | | | |





**Appendix C. Comparison of Regression Models and Normality Tests for May–June Precipitation**

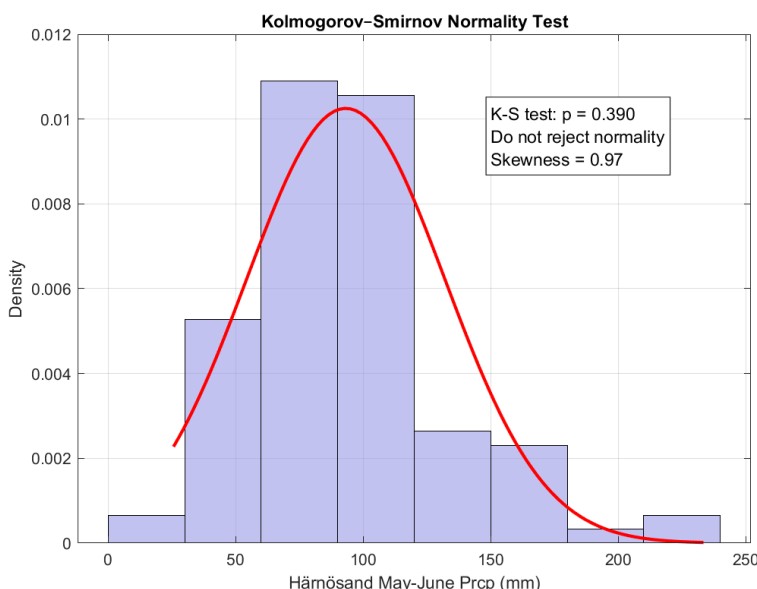

**Figure C1. Distribution of May–June precipitation recorded at the Härnösand weather station for the period 1920–2021, along with the results of the Kolmogorov–Smirnov test for normality.**

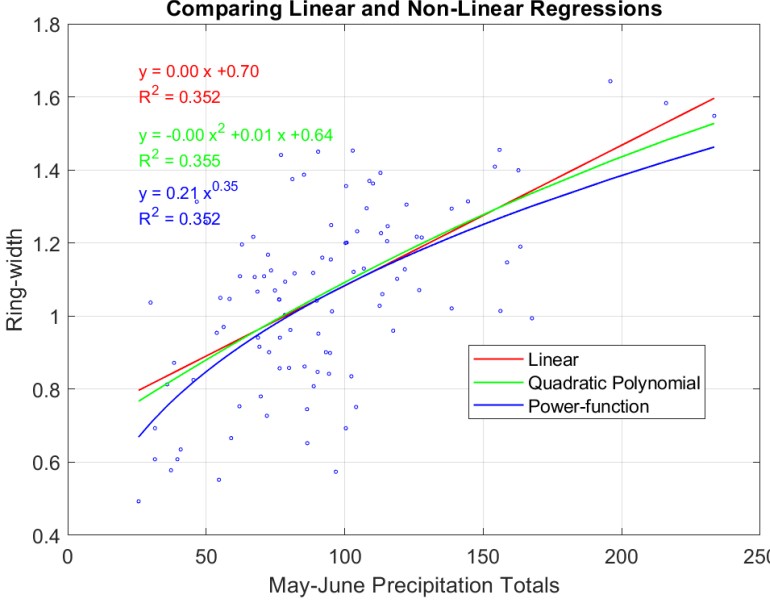

**Figure C2. Scatter-plot of May–June precipitation totals with fitted regression models (linear, polynomial and power function). Corresponding regression equations and $R^2$ values are displayed in the figure.**





## Appendix D. Comparison with other hydroclimatic reconstructions from Fennoscandia

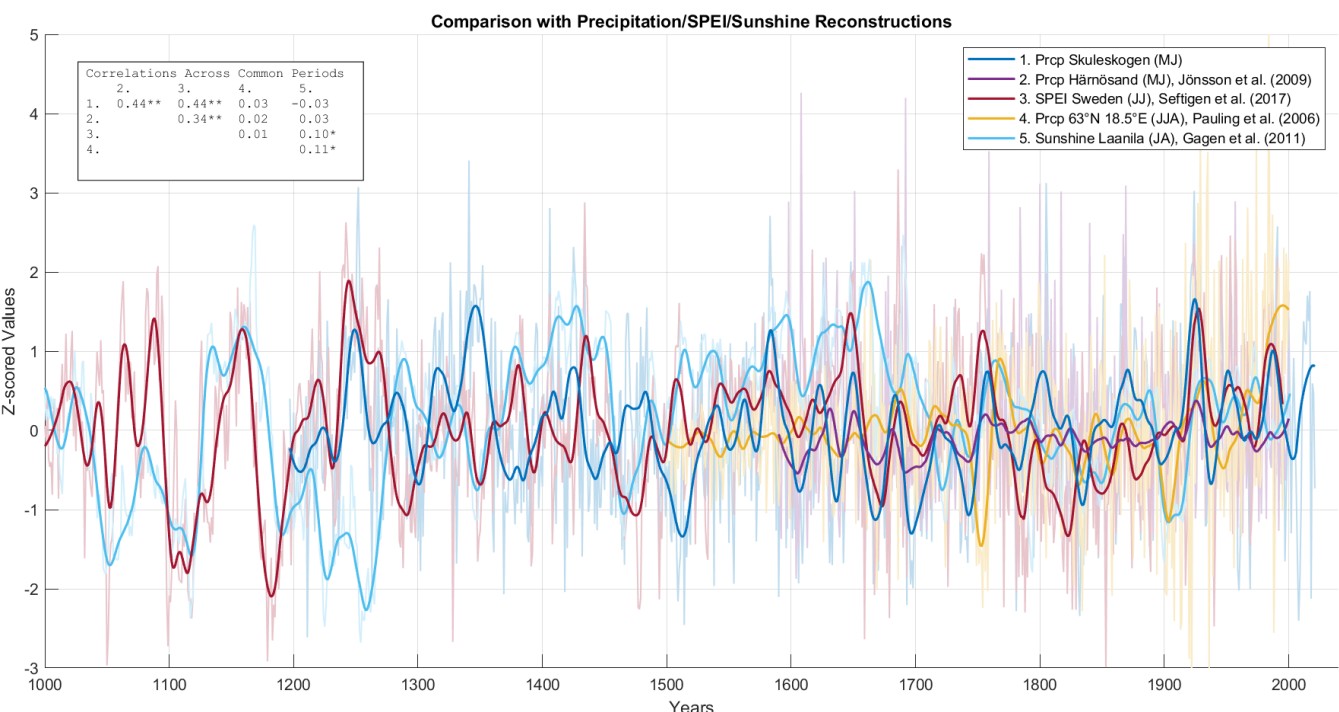

**Figure D1. Comparison of our reconstruction (63° N, 18.5° E) with other hydroclimatic reconstructions in the region, including May–June precipitation near Härnösand (62.6° N, 18.0°E; Jönsson & Nilsson, 2009), June–July drought index (SPEI) for 55°–65° N and 5°–30° E (Seftigen et al., 2017), European gridded June–August precipitation (grid at 63° N, 18.5° E used) (Pauling et al., 2006), and sunshine reconstruction from Lannia, northwest Finland (65.0° N, 25.5° E). All series are standardized to z-scores. Correlation matrix showing Pearson's r across common periods is showed in the upper left corner, \*$P < 0.05$, \*\*$P < 0.01$.**




**Appendix E. Extreme years**

**Table E1.** Years with the greatest deviations from the 28-years spline, expressed as standard deviations, for the period of the chronology with EPS > 0.85.

| Rank | Year (lowest) | Std. Dev. | Year (highest) | Std. Dev. |
|------|---------------|-----------|----------------|-----------|
| 1 | 2018 | −2.95 | 1805 | 2.98 |
| 2 | 1590 | −2.89 | 1406 | 2.88 |
| 3 | 2008 | −2.78 | 1341 | 2.57 |
| 4 | 1418 | −2.75 | 1434 | 2.47 |
| 5 | 1331 | −2.54 | 1583 | 2.44 |
| 6 | 1960 | −2.39 | 1626 | 2.39 |
| 7 | 1538 | −2.31 | 1991 | 2.39 |
| 8 | 1781 | −2.30 | 1929 | 2.29 |
| 9 | 1795 | −2.25 | 1425 | 2.28 |
| 10 | 1328 | −2.22 | 1924 | 2.24 |






## Appendix G. Climate correlation maps for June

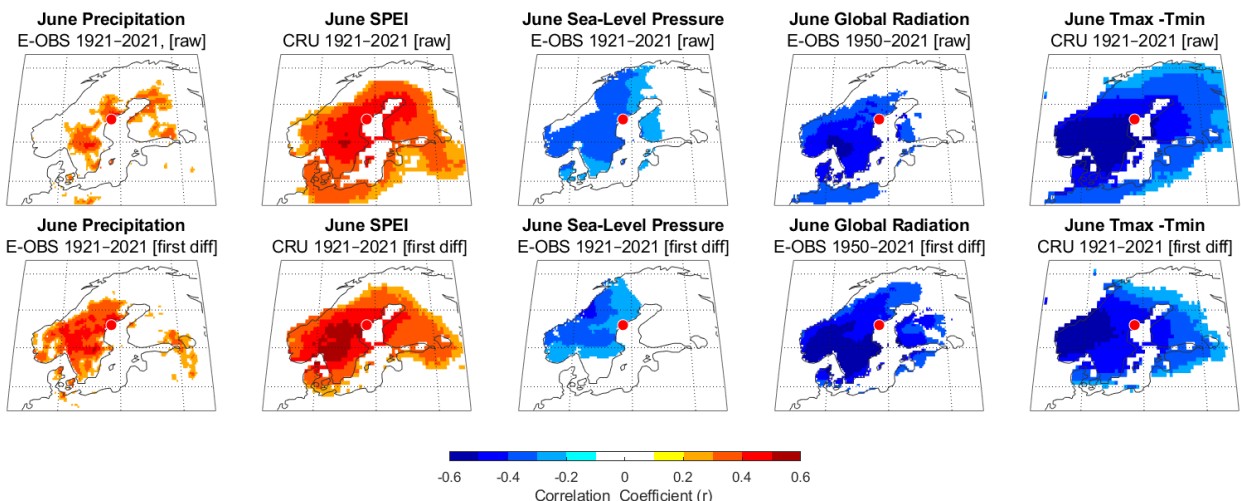

**Figure G1. Correlation maps (Pearson's r values, *P* < 0.01) between tree-ring width chronology and the instrumental data for June over the 1921–2021 period. Raw data (upper panel) and first-difference data (lower panel) are used. The red dot denotes the location of our chronology.**

## Data availability

The precipitation reconstruction will be made available from the National Oceanic and Atmospheric Administration's (NOAA) Paleoclimatology Program/World Data Service for Paleoclimatology: https://www.ncei.noaa.gov/products/paleoclimatology (National Oceanic and Atmospheric Administration's Paleoclimatology Program, 2025).

## Author contributions

**PS**: Writing – original draft, Investigation, Software, Formal analysis, Visualization. **JS**: Writing (review and editing), Funding acquisition, Supervision, Investigation. **FCL**: Writing – original draft, Supervision, Investigation. **JS**: Writing – review & editing, Investigation, Resources. **MF**: Writing – review & editing, Investigation. **PJK**: Writing – review & editing, Formal analysis, Software. **ZL**: Writing – review & editing, Formal analysis, Software. **KS**: Writing – review & editing, Conceptualization, Funding acquisition, Supervision, Investigation.

## Declaration of Competing Interest

The authors declare that they have no known competing financial interests or personal relationships that could have appeared to influence the work reported in this paper.



## Acknowledgements

We thank Tim Brandin, Alexandra Seftigen, and Peter Seftigen for assistance during the fieldwork in Skuleskogen in July 2022.

## Financial support

This work was supported by the Swedish Research Council (Vetenskapsrådet, grant no. 2019-05228 to K.S.), Carl Tryggers Stiftelse för Vetenskaplig Forskning (project no. CTS 21:1469 to K.S), and Adlerbertska Forskningsstiftelsen (funding to K. S.). F.C.L. was supported by the Swedish Research Council (Vetenskapsrådet, grant no. 2023-00605), the Marianne and Marcus Wallenberg Foundation (grant no. MMW 2022-0114), and the Centre for Advanced Study (CAS) at the Norwegian Academy of Science and Letters which funded and hosted the research project "The Nordic Little Ice Age" during the 2024/2025 academic year.

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
