# Peer review of "Seven centuries of rainfall reconstructed from Scots Pine ring width in sub-Arctic Sweden"

_EGUsphere, 2025_

## Referee Comment (RC1)

**Review of Seven centuries of rainfall reconstructed from Scots pine ring width in sub-Arctic Sweden.**

Authors: Petter Stridbeck, Jesper Björklund, Fredrik Charpentier Ljungqvist, Jennie Sandström, Mauricio Fuentes, Paul J. Krusic, Zhi-Bo Li, and Kristina Seftigen

This is a solid and carefully prepared manuscript that presents the longest and northernmost treering-based precipitation reconstruction from Sweden. The work contributes valuable new data to Scandinavian hydroclimatology. The methodology is rigorous, and the discussion is broad and well connected to previous studies.

The main strengths are:

- Long and well-dated chronology
- Thorough methodological approach using mRCS and validation statistics
- Clear comparison with previous reconstructions
- Well-documented site and regional context

This is a valuable and well-executed contribution. The manuscript would be substantially improved by a clearer statement of objectives, tighter structure, explicit figure references, and more careful separation of data-supported results from speculative interpretations. After these major revisions, the paper will be suitable for publication.

**General**

**Length and structure**

The manuscript is rather long for a paper of this type, spanning 36 pages including extensive appendices and detailed methodological descriptions. While the length reflects the thoroughness of the work, the readability could benefit from some condensation. I suggest that the authors consider shortening or restructuring parts of the introduction and discussion to improve focus and flow, and moving some of the more technical details (for example the full description of the mRCS procedure) to the Supplementary Material. This would help highlight the key results and interpretations without compromising the study's scientific depth.

**Introduction and aims**

The introduction provides a clear and well-structured background and rationale for the study, but the specific research questions or objectives are not stated explicitly. I recommend adding a short paragraph at the end of the introduction (after line 91, ending with "...relied on living trees.") that clearly formulates the main aims of the paper, for example:

"The specific objectives of this study are to (1) assess whether Scots pine (Pinus sylvestris L.) growing under drought-stressed conditions at the High Coast in northern Sweden record a robust precipitation signal suitable for hydroclimate reconstruction; (2) develop and evaluate a May–June precipitation reconstruction extending as far back as possible using both living and subfossil material; (3) analyse the temporal stability and spatial extent of the climate–growth relationships to determine the regional representativeness of the signal; and (4) identify possible multidecadal cycles and compare the reconstructed variability with other Scandinavian and European hydroclimate records."

The main research questions are all addressed in the discussion, though not explicitly referenced back to the aims. It would strengthen the paper if the authors clearly link each subsection of the discussion to the corresponding objectives, ensuring closure on each question.

Precipitation data and spatial heterogeneity

Because precipitation is inherently heterogeneous, its spatial coherence is much lower than that of temperature. Consequently, gridded datasets such as CRU TS or E-OBS, with resolutions of 0.5° and 0.1°, may not capture local-scale variability relevant for a coastal and topographically complex area like the High Coast. The weaker correlations between the tree-ring chronology and CRU precipitation compared with the local Härnösand station data likely reflect this limitation.

It also remains unclear which meteorological stations underpin the gridded datasets in this region. A brief inspection suggests that the CRU grid cell covering 62.5–63.5° N, 18–19° E may include data from both Norwegian and Finnish stations, introducing potential biases due to differing precipitation regimes. I recommend that the authors clarify which stations contribute to the grid, or at least acknowledge that the cell likely averages over multiple climatic zones. A short discussion noting that locally observed data (Härnösand) provide a more representative calibration target for hydroclimatic reconstruction at this spatial scale would strengthen the argument and contextualize the correlation differences between datasets.

**Specific**

**Figures 1–11**

Not all figures are explicitly referenced or discussed in the main text. Each figure should be clearly cited at least once in the narrative, ideally where the corresponding result is described. For instance, Figure 10 is not mentioned in Section 4.3 even though the discussion refers to the spectral features it presents, and Figure 9 is only indirectly discussed. Consistent figure referencing is essential for clarity and for guiding the reader through the results. I recommend that the authors carefully check that all figures are cited and briefly explained in the text, ensuring that their relevance to the argument is clear.

**Section 4.3**

Line 464 ff European harvest failures and drought sensitivity

The discussion suggests that periods of reduced growth in the Skuleskogen chronology coincided with major European harvest failures (for example the 1690s famine). Given that the reconstruction represents May—June precipitation variability from a single site in northern Sweden, and that precipitation is spatially heterogeneous, such continent-wide connections should be expressed with more caution. I recommend that the authors rephrase this passage to emphasize that any apparent temporal overlap with European harvest crises is coincidental rather than implying a direct climatic correspondence.

The statement "This emphasises the drought-sensitivity of the Skuleskogen tree-ring chronology" (L467) is ambiguous. The preceding sentences describe enhanced growth during periods of reduced solar activity and cooler conditions, which would imply higher moisture availability rather than actual drought. I recommend rephrasing to clarify that the chronology reflects tree growth being limited by moisture (that is increased growth under wetter conditions), rather than suggesting that these cool periods themselves demonstrate "drought sensitivity."

**Lines 474 ff Comparison with 1867**

The comparison between the 1690s and the 1867 harvest failure reads as speculative. The argument that the lack of a growth minimum in 1867 "highlights the exceptional nature" of the 1700 CE minimum is not clearly supported by data. It is uncertain whether the absence of a signal in 1867 reflects genuinely different climatic conditions (for example wetter weather) or simply that tree growth at Skuleskogen is insensitive to temperature-driven short growing seasons. I suggest softening this interpretation and acknowledging the speculative nature of this comparison.

**Lines 481 ff Figure 10 and spectral interpretation**

The paragraph discussing the multidecadal oscillations appears to refer to Figure 10, but the figure is never explicitly cited. Every figure presented should be clearly referenced in the text, and its relevance briefly explained. The text should explicitly direct the reader to Figure 10 and describe what it shows (the 34- and 64-year peaks and their temporal variation).

Moreover, the argument that these periodicities correspond to large-scale modes such as the AMO or PDO is not convincing. The comparison is based solely on similar timescales, without any statistical or mechanistic evidence of linkage. Unless quantitative support (for example correlation or coherence analysis) is provided, I recommend removing the speculative reference to the AMO and PDO altogether.

**Section 4.4 Undated material**

Section 4.4 discusses the presence of numerous undated deadwood samples and suggests that some may predate the established chronology, implying potential for further extension. However, the text does not specify the proportion of material that remained undated or excluded. For readers to evaluate the true potential for extending the chronology, it would be important to quantify this, for example by stating the number (or approximate percentage) of collected samples that could not be cross-dated, and whether these were excluded due to poor preservation, missing sapwood, or non-overlap with the master chronology.

---

## Author Comment (AC1)

*General Response to the Editor and Reviewers*

A few recurring points were raised during the review process: (i) the manuscript was considered somewhat lengthy and would benefit from trimming, (ii) some figures required improved clarity (iii) the discussion of potential large-scale connections between the identified cycles was regarded as overly speculative and better be toned down or removed.

In response, we have streamlined the text where possible, revised the figures to enhance their clarity, and substantially moderated the discussion of large-scale cyclic connections. While one reviewer suggested that this could be developed further, we selected not to make additional analysis in the manuscript, which is already, as pointed out, rather extensive. In this version of the manuscript, we simply present pre-existing power spectra and explicitly state the periods over which the two cycles are most dominant, without further speculative interpretation.

**Point-by-point Response to Reviewer 2:**

**R2.1**

*References. The manuscript would benefit from including a few key recent works, especially those related to long-term hydroclimate variability and cyclic patterns. I recommend citing in the introduction Esper et al. (2024), and when talking about microsites, Hartl et al. (2021). Additionally, the recent work done by Torbenson et al. (2025), could provide relevant methodological and interpretive context for multidecadal variability and the role of large-scale ocean–atmosphere dynamics, particularly regarding the Maunder Minimum..*

**Comment:** Esper et al have conducted important work in the region that should be cited, and micro site condition by Hartl et al is highly relevant for our work. Both are now cited in the introduction (L39/L46). The work by Torbenson et at are brought up in the discussion related to the Maunder Minimum (L451).

**R2.2**

*Figure 9 – This is arguably the main figure of the paper, yet its current design is suboptimal. The reconstruction, uncertainty bands, and historical annotations are visually cluttered. I suggest a clearer graphical layout ents. Lower the shades and highlight the reconstruction..*

**Comment:** Efforts have been made to make this figure less cluttered, without removing relevant information, such as removing crossing lines and some text. We trust it is found to be in an acceptable state after these adjustments.

**R2.3**

*Discussion of cyclicity and solar forcing – The discussion of the ~64-year and ~34-year cycles is intriguing but could be developed further. In particular, the authors briefly mention the Dalton and Spörer minima but omit the **Maunder Minimum**, which is essential in the context of multi-decadal hydroclimate variability and possible solar forcing. I recommend adding a more explicit discussion of the Maunder Minimum, potential links to solar cycles (e.g., Gleissberg or Suess/de Vries), and whether these correspond to the periodicities found in the reconstruction and in AMO/PDO.*

**Comment:** For reasons expressed in the general section at the top of this document we have chosen not to go deeper into the possible causes of the observed periodicities identified in the time series.

We agree that, particularly as we bring up both Spöer and Dalton, the maunder minimum should not be neglected, and is added as an annotation in figure 9. Further, we emphasize that this period constitutes an exception to the overall tendencies observed throughout the rest of the chronology (i.e., cold conditions tends to be favorable for tree growth at site) (L444f). We further note that the short cycle reaches its maximum strength in the wavelet analysis during the Maunder Minimum, without speculating on the underlying causes (L446f).

**R2.4**

*Appendix Figure D1. Very difficult to interpret. I suggest to include a larger plot with pannels for each reconstruction. Perhaps use only the 11 y running mean with shades in red for dry, and blue for wet periods. I would be better for interpreting all the figures. Otherwise it is useless..*

**Comment:** We agree that this figure clearly was suboptimal and it has now been changed into a stackplot, which makes it considerable easier to discern the presented time series. The correlation matrix in the top of the figure is also informative.

**R2.5 Minor edits**

*Line 39 add Esper et al., 2024.. Difficult to explain why it is ommitted.. Esper, J., Torbenson, M. & Büntgen, U. 2023 summer warmth unparalleled over the past 2,000 years. Nature 631, 94–97 (2024). https://doi.org/10.1038/s41586-024-07512-y*

**Comment**: Added at L39

*Line 46. Include https://doi.org/10.1016/j.dendro.2020.125787*

**Comment**: Hartl et al added at L46

*Line 164, (march??) what´s that?*

**Comment**: Changed to mRCS

*Line 312, what about comparing it with the OWDA or the most recently updated GEDA? that would be more sound than Pauling.. and apples to apples..*

**Comment**: I is a valid point to included GEDA in the comparison, and this is now included Appendix Figure D1.

*Line 465.. and the Maunder?? it is the solar minima of the last 500 years.. why is it not mentioned? you can discuss such growth-climate disruption using the recently published Torberson et al., 2025. Torbenson, M. C. A., Stahle, D. W., Cook, E. R., Cook, B. I., Büntgen, U., Chen, F., et al. (2025). Disruption of drought teleconnections between ENSO-influenced regions around 1700 CE. Geophysical Research Letters, 52, e2025GL115600. https://doi.org/10.1029/2025GL115600*

**Comment**: Maunder minimum is now discussed and included in the main figure, see further comments regarding this in R2.3

*Lines 485-490. More discussion on the relationship of AMO/ PDO and the positive and negative cycles.. is there a connection??? this is very important and could add value to the reconstruction.*

**Comment**: The possible relationship with AMO/PDO has been removed in this version of the manuscript as discussed in the introduction.

---

## Author Comment (AC2)

**General Response to the Editor and Reviewers**

A few recurring points were raised during the review process: (i) the manuscript was considered somewhat lengthy and would benefit from trimming, (ii) some figures required improved clarity (iii) the discussion of potential large-scale connections between the identified cycles was regarded as overly speculative and better be toned down or removed.

In response, we have streamlined the text where possible, revised the figures to enhance their clarity, and substantially moderated the discussion of large-scale cyclic connections. While one reviewer suggested that this could be developed further, we selected not to make additional analysis in the manuscript, which is already, as pointed out, rather extensive. In this version of the manuscript, we simply present pre-existing power spectra and explicitly state the periods over which the two cycles are most dominant, without further speculative interpretation.

**Point-by-point Response to Reviewer 1:**

**R1.1**

*The introduction provides a clear and well-structured background and rationale for the study, but the specific research questions or objectives are not stated explicitly. I recommend adding a short paragraph at the end of the introduction (after line 91, ending with "...relied on living trees.") that clearly formulates the main aims of the paper, for example:*

*"The specific objectives of this study are to (1) assess whether Scots pine (Pinus sylvestris L.) growing under drought-stressed conditions at the High Coast in northern Sweden record a robust precipitation signal suitable for hydroclimate reconstruction; (2) develop and evaluate a May–June precipitation reconstruction extending as far back as possible using both living and subfossil material; (3) analyse the temporal stability and spatial extent of the climate–growth relationships to determine the regional representativeness of the signal; and (4) identify possible multidecadal cycles and compare the reconstructed variability with other Scandinavian and European hydroclimate records."*

**Comment:** Thanks for pointing this out as well as suggestions for wording. A slightly modified version is added at the end of the introduction (Line 83ff)

**R1.2**

*Because precipitation is inherently heterogeneous, its spatial coherence is much lower than that of temperature. Consequently, gridded datasets such as CRU TS or E-OBS, with resolutions of 0.5° and 0.1°, may not capture local-scale variability relevant for a coastal and topographically complex area like the High Coast. The weaker correlations between the tree-ring chronology and CRU precipitation compared with the local Härnösand station data likely reflect this limitation.*

*It also remains unclear which meteorological stations underpin the gridded datasets in this region. A brief inspection suggests that the CRU grid cell covering 62.5–63.5° N, 18–19° E may include data from both Norwegian and Finnish stations, introducing potential biases due to differing precipitation regimes. I recommend that the authors clarify which stations contribute to the grid, or at least acknowledge that the cell likely averages over multiple climatic zones. A short discussion noting that locally observed data (Härnösand) provide a more representative calibration target for hydroclimatic reconstruction at this spatial scale would strengthen the argument and contextualize the correlation differences between datasets.*

**Comment:** This is an important comment, the differences between the different data sets are now discussed more throughout, and why it is not unexpected the CRU data capture local variability in precipitation to less extent than the other data sets we have used (Line 408ff).

Note that we used average data from CRU TS and E-OBS for the same spatial region (62.5–63.5° N, 18–19° E), we for precipitation observed substantially stronger correlation with the chronology for the E-OBS data (Figure 5 & 6) and also was more similar to the local station Härnösand.

Further, the reviewer raises the question of which stations can be expected to contribute to the grid. From reading the literature (e.g. Harris et al 2020) and scrutinizing the distance-decay weighting function applied in the CRU dataset it appears that the reviewer's statement indeed could be correct. However, we consider a detailed assessment of the relative contribution of individual stations to be of limited additional value and have not attempted, and we hope discussion it is considered to be satisfactory.

**R1.3**

*Not all figures are explicitly referenced or discussed in the main text. Each figure should be clearly cited at least once in the narrative, ideally where the corresponding result is described. For instance, Figure 10 is not mentioned in Section 4.3 even though the discussion refers to the spectral features it presents, and Figure 9 is only indirectly discussed. Consistent figure referencing is essential for clarity and for guiding the reader through the results. I recommend that the authors carefully check that all figures are cited and briefly explained in the text, ensuring that their relevance to the argument is clear.*

**Comment:** Thank the reviewer for pointing this out. Figure 9 is directly cited starting at line 285. Other figure citation previously missing: Figure 1 (L141), figure 10a and 10b is cited in section 4.3 at (L449/451), and Appendix E (L337).

**R1.4**

*The discussion suggests that periods of reduced growth in the Skuleskogen chronology coincided with major European harvest failures (for example the 1690s famine). Given that the reconstruction represents May–June precipitation variability from a single site in northern Sweden, and that precipitation is spatially heterogeneous, such continent-wide connections should be expressed with more caution. I recommend that the authors rephrase this passage to emphasize that any apparent temporal overlap with European harvest crises is coincidental rather than implying a direct climatic correspondence.*

**Comment:** We now explicitly state that the annotations in figure 9 illustrate temporal associations only and do not imply a necessary or demonstrable causal relationship (line 286f). Nevertheless, we consider it of interest to place the hydroclimatic conditions in the north indicated by our chronology in the context of extreme and climate-related events across Europe. Section 4.3 begins by discussing these potential associations (L431ff).

**R1.5**

*The statement "This emphasises the drought-sensitivity of the Skuleskogen tree-ring chronology" (L467) is ambiguous. The preceding sentences describe enhanced growth during periods of reduced solar activity and cooler conditions, which would imply higher moisture availability rather than actual drought. I recommend rephrasing to clarify that the chronology reflects tree growth being limited by moisture (that is increased growth under wetter conditions), rather than suggesting that these cool periods themselves demonstrate "drought sensitivity."*

**Comment:** We agree that the wording was ambiguous and have revised it to clarify that it indicates that tree growth is "moisture-limited" rather than "drought-sensitive" in this context. (L440)

**R1.6**

*The comparison between the 1690s and the 1867 harvest failure reads as speculative. The argument that the lack of a growth minimum in 1867 "highlights the exceptional nature" of the 1700 CE minimum is not clearly supported by data. It is uncertain whether the absence of a signal in 1867 reflects genuinely different climatic conditions (for example wetter weather) or simply that tree growth at Skuleskogen is insensitive to temperature-driven short growing seasons. I suggest softening this interpretation and acknowledging the speculative nature of this comparison.*

**Comment:** We agree with the reviewer that the logic of our original interpretation was problematic. The reasoning that the inverted growth during the this reflected conditions outside the calibration range, implying a temporary shift to temperature limited growth, cannot be supported by comparison with individual extreme years, and certainly not to 1867 which might have been even colder in the region. We thank the reviewer for pointing this out and have revised the interpretation accordingly. These statements are removed and instead we underlines that this period coincides with the Maunder minimum (L444). Rather, what 1867 shows us (which had an exceptional cold spring and short summer in northern Sweden), is that even during very cold conditions growth can be favorable  as noted at line 442.

**R1.7**

*The paragraph discussing the multidecadal oscillations appears to refer to Figure 10, but the figure is never explicitly cited. Every figure presented should be clearly referenced in the text, and its relevance briefly explained. The text should explicitly direct the reader to Figure 10 and describe what it shows (the 34- and 64-year peaks and their temporal variation).*

**Comment:** Figure 10a and 10b is now cited in the text (L289).

**R1.8**

*Moreover, the argument that these periodicities correspond to large-scale modes such as the AMO or PDO is not convincing. The comparison is based solely on similar timescales, without any statistical or mechanistic evidence of linkage. Unless quantitative support (for example correlation or coherence analysis) is provided, I recommend removing the speculative reference to the AMO and PDO altogether.*

**Comment:** We have removed this from possible connections to AMO / PDO from the paper.

**R1.9**

*Section 4.4 discusses the presence of numerous undated deadwood samples and suggests that some may predate the established chronology, implying potential for further extension. However, the text does not specify the proportion of material that remained undated or excluded. For readers to evaluate the true potential for extending the chronology, it would be important to quantify this, for example by stating the number (or approximate percentage) of collected samples that could not be cross-dated, and whether these were excluded due to poor preservation, missing sapwood, or nonoverlap with the master chronology.*

**Comment:** We have included some detailed information regarding material which have not been incorporated in chronology but potentially could extend it (L471ff)